# Design of a Tribotester Based on Non-Contact Displacement Measurements

**DOI:** 10.3390/mi10110748

**Published:** 2019-10-31

**Authors:** Chang-Lae Kim, Yoon-Gyung Sung

**Affiliations:** Department of Mechanical Engineering, Chosun University, Gwangju 61452, Korea; kimcl@chosun.ac.kr

**Keywords:** displacement, friction, non-contact, tribotester, wear

## Abstract

A tribotester with an integrated load sensor based on a strain gauge is typically used to measure the friction coefficient generated by the contact-related sliding motion of two objects. Since the friction coefficient is obtained by dividing the measured friction force by the applied normal force, the normal and friction forces must be measured for accurate analysis. In this study, a tribotester was used to measure the displacement of a cantilever tip using the fiberoptic sensor in a non-contact method. The friction coefficient measurement using the fiberoptic sensor was proven to be valid by calibrating the tip displacement due to normal/friction forces after designing a basic structural cantilever tip based on experiments and simulation analyses. The results obtained by using the fiberoptic sensor-cantilever tip-based tribotester were compared with those obtained using commercial and/or custom-built tribotesters under the same conditions. By designing various shapes of cantilever tips and using simulation analysis, the calibrations of the normal/friction forces and tip displacement could be verified and the coupling effect was evaluated. The performance and reliability of the fiberoptic sensor-cantilever tip-based tribotester, which can be used to determine the normal/friction forces by non-contact displacement measurements without a strain gauge, were verified.

## 1. Introduction

Problems related to friction and wear due to the relative sliding motion in the contact areas of the driving parts of mechanical equipment are increasing in the machine and material field. Due to friction/wear problems have a large impact on the efficiency, reliability, and durability of the system, lubricants or wear-resistant coatings are generally used to reduce the friction and protect the original material surface [1,2,3,4,5]. Various studies have been conducted to reduce friction and wear phenomena in the contact area and the degree of the improvement of the friction properties is evaluated using diverse methods [6,7,8,9,10,11]. The friction force generated by the contact of two objects is measured and the friction coefficient, which is the friction force divided by the applied normal force, is used to evaluate the friction properties of the material. The friction coefficient remains constant because the increase in the friction force is proportional to the increase in the normal force. However, it has been reported that the friction coefficient varies depending on the applied normal force, especially in nanoscale experiments [12,13]. When the scale is small, there is the possibility that a minute change can affect the friction properties and the friction coefficient (tendency to frictional variation) may differ because the tendency of wear changes depending on the applied normal force.

It is essential to determine the contact pressure conditions when conducting a frictional experiment [14,15]. The contact pressure varies depending on the applied normal force and contact area. The contact pressure generally increases with decreasing contact area, while the applied normal force increases. If the load on the actual system is very large (tens of N to tens of kN), it is not easy to install a large system or apply large loads in each experiment. Although the applied load is small, the contact pressure can be increased by reducing the contact area. The experiment is conducted by assuming that the interface between two objects in contact is the point of contact because the surface contains small asperities, although the actual system is in a state of area contact [16,17,18]. The experiment can also be conducted in the point contact state using a ball or pin (acceleration experiment). The wear phenomenon may be accelerated because the contact pressure substantially increases. With respect to ball contact, heterogeneous contact induced by a tilt phenomenon, which can occur during line or area contact, can be prevented. 

In an actual system, the method used to evaluate the friction properties, sliding/rolling motion state, point/line/area contact state, reciprocation/rotation state, and other states, can be changed based on the state of the contact [19,20,21]. In most studies, experimental methods depend on contact conditions in situations similar to actual applications. Many research teams use commercial tribotesters and self-designed experimental equipment to evaluate the friction/abrasion properties [6,7,8,9,10,11,22,23]. A strain gauge-based load sensor is generally used to measure the friction force. In addition to the electrical resistance, the voltage is changed under constant current because of the change in the resistance when the cantilever of the load sensor with the strain gauge is distorted by friction. The voltage is calibrated using the force (N). In other words, the physical friction phenomenon is first changed into an electrical signal using the strain gauge and then transformed to the friction force, which is used to compare and evaluate the physical phenomenon.

The friction properties can be accurately analyzed when the friction force is measured in real time. The types and ranges of friction force sensors depend on the size of the experimental device. On the other hand, in the case of normal force, the following two methods are commonly used. First, the dead weight method is traditionally used to apply a normal force by using a weight with a known mass. This method has been established when the tribotester was initially developed; however, it is still used. Nowadays, a target weight is applied by moving the displacement stage immediately after the contact of two objects (tip and specimen) [6]. The correlation between the travel distance and weight can be determined by calibrating the normal force according to the traveled distance from the moment of contact using an electronic scale. The method based on which the normal force is manually applied is similar to the dead weight method. In addition, the force is automatically applied using a step-motorized stage in the vertical direction. In this case, the vertical displacement is automatically adjusted to maintain a constant normal force throughout the experiment.

The methods used to measure the normal and friction forces with a universal strain gauge have few limitations. First, the friction force sensor detects the degree of change of the cantilever of the load sensor in the lateral direction (direction of friction force generation). The strain gauge attached to the cantilever is already deformed in the vertical direction (direction of the normal force), even before the experiment, because of the normal force that is transferred to the cantilever. Therefore, there is a high possibility that the data measured by the deformed strain gauge are erroneous and the value obtained through calibration and the actual friction value may differ. This coupling effect occurs due to the interruption of the normal and friction forces [24,25]. This effect is negligible in a large-scale experiment because it is based on a trivial error; however, it has to be addressed when dealing with micro/nanoscale friction forces. Due to the specification of the strain gauge which can measure the force in the lateral direction, a limited amount of load must be applied in the vertical direction. This means that the normal force conditions in the experiment are limited. The scope of the normal and friction forces must be predetermined during the design of the experimental equipment. In other words, when a large load is measured through a strain gauge having a small range of measurable force, it is difficult to obtain an accurate force value because it is out of the measurable range of the strain gauge, or may cause breakage of the strain gauge. On the other hand, when a strain gauge with a large force range is used, it is difficult to measure small changes in the friction or normal forces because the resolution of the minimum force is worse and it is difficult to obtain accurate data, especially in microscale fine-precision friction tests. In addition, environment factors, such as vibrations or electrical noise generated by the experimental equipment, temperature, and humidity, also have an influence.

In this study, the calibration of a force unit by measuring the displacement change of a cantilever is proposed instead of normal/friction force measurements using a strain gauge-based load sensor. The degree of deformation of the cantilever tip by normal/friction forces is determined using a fiberoptic sensor, which measures the distance difference based on a non-contact method. The displacement change due to normal and friction forces is determined using displacement-force calibration. This method solves problems related to the strain gauge. The reliability of the calibrations of the normal/friction forces using this device is proven using the experiment and finite element analysis (FEA). The normal/friction forces were measured using a simple friction experiment with a custom-built experimental device. Subsequently, the friction coefficient and wear track were analyzed. The results were compared with those measured with commercial/custom-built friction testing devices that use strain gauge-based load sensors. By designing cantilever tips with different shapes, the coupling effect generated by normal/friction forces and their interference were analyzed using FEA simulations.

## 2. Experimental Details

### 2.1. Fiberoptic Sensor [26,27,28] 

A fiberoptic sensor (RC100, PHILTEC, Inc., Annapolis, MD, USA), which can be used to measure the distance, displacement, and vibration, consists of two parts, that is, a cable part and probe part. This sensor has a reflectance-compensated output, spot size of 2.5 mm, maximum operating range of 5 mm, and sensitivity of 1.3 mV/µm. Figure 1a shows the schematic design explaining the operation principle and a picture of the fiberoptic sensor. The external size of this sensor is shown in the picture. The light from the fiberoptic probe is emitted in a straight line and received by the light detector after reflection by a target (mirror). The light intensity is used to estimate some distance away and the correlation between the voltage and distance can be matched by simple calibration before its use. The actual change in the distance and sensing distance can be matched by the simple installation of the experimental device for displacement calibration, as shown in Figure 1b. Two fiberoptic sensors were calibrated because the normal and friction forces are simultaneously measured in this study. The relationship between the actual change in the distance and displacement measured by the sensor is linear and the numerical values are reasonable.

### 2.2. Experimental Setup

The reciprocating-type tribotester for the sliding method, which comprises fiberoptic sensors, was custom-built. Two fiberoptic sensors were installed. A step-motorized stage (5-phase stepping motor-RK, VEXTA, Japan) was used, which allows reciprocating motion with precise displacement. The driving stage with a maximum travel distance of 90 mm and speed of 20 mm/s can modulate the reciprocating distance, speed, and time because it is connected to the central processing unit (CPU) through a controller and driver. The moving displacement of the stage can be determined in real time with a monitor and the normal and friction forces are determined by measurements with fiberoptic sensors. The schematic design of the custom-built experiment jig with fiberoptic sensors is shown in Figure 2a and images of the jig and the cantilever tip are shown in Figure 2b. The stainless steel cantilever tip is 33 mm long, 25 mm wide, and 0.65 mm thick. As shown in the images, the width of both arms of the cantilever tip in which the displacement change occurs in the vertical direction is 0.35 mm. The light emitted from the fiberoptic probe is reflected on the cantilever tip or mirror. As the reflected light returns back to the fiberoptic sensor, the distance change is measured. The distance changes measured in the vertical and lateral directions represent the degree of displacement of the tip due to normal and friction forces, respectively.

### 2.3. Sensor Calibration

During the calibration of the fiberoptic sensors, the displacement value is first calibrated using the displacement of the cantilever tip. Subsequently, a calibration is carried out to match the force to the displacement of the sensor. The force calibration is the process that is used to match the degree of deformation to the force (N) using the stiffness of the cantilever tip. The relationship between the displacement changes of the cantilever tip in the vertical and lateral directions and the forces, that is, the normal force and friction forces, based on the calibration process is linear. From the results of the calibration to analyze the relationship between the applied force and the displacement of the cantilever tip, it is confirmed that the cantilever tip stiffness (3,950 N/m) in the lateral direction is approximately five times larger than the cantilever tip stiffness (710 N/m) in the vertical direction. The stiffness of the cantilever tip can be calculated using Equation (1) [29,30]:(1)$$S=f\;\left(N\right)/d\;\left(m\right)$$
where *S* is the stiffness of the cantilever tip, and *d* and *f* are the displacement and force generated in the vertical and lateral directions, respectively.

An additional calibration of the cantilever tip and the fiberoptic sensor was conducted using finite element analysis (FEA) simulations (ABAQUS, Dassault Systèmes, Vélizy-Villacoublay, France), as shown in Figure 3. A wider range of calibrations was carried out by extending the measurement range. Then, it was compared with the calibration values actually measured throughout the experiment. The calibration results generally show a linear relationship, confirming the reliability of the cantilever tip of the fiberoptic sensor. Based on this process, it is possible to obtain the values of the normal and friction forces in the experiment.

### 2.4. Experiment Conditions

Friction and wear tests were conducted using the fiberoptic sensor-cantilever tip-based tribotester (Figure 2 and Figure 4a). A zirconia ball with a diameter of 1 mm is attached to the end of the cantilever tip and used as a contact pin. To evaluate the performance and reliability of the test apparatus, a silicon wafer substrate with low surface roughness and overall uniformity was used as a specimen. With a load of 10 mN (immediately after the cantilever tip was brought into contact with the specimen, the stage was moved down in the vertical direction such that a displacement change equivalent to 10 mN occurred), the relative sliding reciprocating motion between the tip and the specimen proceeded at a speed of 4 mm/s. The sliding stroke was 2 mm and the experiments were performed up to a total sliding distance of 250 mm. The generated friction force and the normal force applied during the experiment were measured in real time. The friction coefficient was obtained by dividing the measured friction force by the normal force, as shown in Equation (2) [31]:(2)$$\mu =\;Ff\;\left(N\right)/Nf\;\left(N\right)$$
where *µ* is the dimensionless friction coefficient, *Ff* is the friction force, and *Nf* is the normal force.

The friction coefficient obtained by the tribotester comprising the fiberoptic sensors was compared with the values measured by tribotesters using the load sensor and conventional strain gauge. A commercial tribotester (UMT-2, CETR, Campbell, CA, USA) and custom-built micro-tribotester were used in this study. In the commercial tribotester (Figure 4b), the stage that moves in the vertical and lateral directions is automatically adjusted and the forces (normal and friction forces) in both directions are measured in real time by the load sensor. The range of the forces is ~1 mN–200 N and the stage speed is 0.001–10 mm/s. The maximum travel is ~150 mm and the displacement resolution is 1 µm. In the custom-built micro-tribotester (Figure 4c), the friction force in the lateral direction is measured using the load sensor and the normal force is manually applied by the displacement change according to the displacement-force calibration. Therefore, the friction force is measured in real time, but the initially applied load is regarded to be sustained when it comes to the normal force. The load sensor can measure forces up to 50 mN and the resolution of the nonlinearity, hysteresis, and repeatability is 0.5%. The stage is operated by converting the rotary motion of the motor into a reciprocating rectilinear motion using a camshaft connected to the motor and the sliding travel can be adjusted by modulating the radius of the camshaft. All experiments were conducted under the same conditions as those used for the tribotester with the fiberoptic sensor and cantilever tip.

## 3. Results

### 3.1. Friction and Wear Test

As shown in Figure 5a, the tribotester based on the fiberoptic sensor measures the displacement of the cantilever tip in the vertical direction in real time. The 10 mN normal force, which is applied according to the displacement-force calibration, slightly and continuously changes during the friction experiment. The normal force changes because the cantilever tip moves in the vertical direction as wear occurs on the surface of the specimen and vibration occurs due to the driving mechanism of the reciprocating motion in the lateral direction and harshed surface roughness. In the commercial tribotester CETR, when the applied normal force is out of the range of 10 mN, the stage in the vertical direction moves automatically and adjusts the normal force value of 10 mN. The change in the normal force can be measured in real time with this device. In the case of the custom-built micro-tribotester, the 10 mN normal force is applied using the stiffness of the suspension tip; however, it is assumed that the normal force remains constant during the friction experiment because a sensor for the measurement of the change in the normal force in real time is not installed. Figure 5b–c show the friction coefficients measured using three different types of tribotesters. The friction coefficient change according to the total sliding distance and the average friction coefficient were determined. The average value and standard deviation were calculated after conducting three or more repeated experiments under the same conditions. The friction coefficient changes are generally similar, especially those (0.2–0.4) of the fiberoptic sensor-cantilever tip-based and commercial CETR tribotesters. The overall friction coefficient change of the custom-built micro-tribotester with a load sensor is similar; however, but significant differences is observed in some sections (initial state to 100 mm sliding distance) and some datapoints stand out. This difference in frictional coefficient is the extent to which it may occur with other experimental devices. The standard deviation is larger than those obtained in the two other experiments, mainly because the real-time change in the normal force is not considered. Because tribological phenomena, in which the influence of the surface roughness and wear is not negligible, are very complex, the generation of completely different friction properties is possible [32]. The reliability of the measuring device must be proven to accurately analyze the friction mechanism. The results in the previous studies show that the frictional characteristics differ depending on the contact conditions and that these variables affect the measurement of the normal or friction forces in micro/nano-scale experiments. The average friction coefficients obtained for the fiberoptic sensor-cantilever tip-based and commercial CETR tribotesters are similar (0.41). The average friction coefficient of the custom-built micro-tribotester is slightly higher (~0.44). The friction coefficient in the section with a sliding distance of ≤100 mm is more than 0.5 higher. The main cause for this might be that the device is not finely modulated with respect to the normal force. The load sensor that measures the friction force in the lateral direction is deformed by the normal force in the vertical direction because the normal force is applied by the vertical displacement of the cantilever tip, which is directly connected to the load sensor. The friction coefficient value could be affected by this coupling effect. It can be confirmed that the average friction coefficient differs depending on the device used for the experiment, although the difference is insignificant. In contrast, only slight differences occur when the custom-built micro-tribotester with a load sensor is used because this system does not monitor the normal force in real time. The experimental results are reliable because the difference is small.

Figure 6 shows the results of the analysis of the wear tracks, which are generated during the friction experiments using three different experimental devices, measured by a scanning electron microscope (SEM, JSM-6610, JEOL Inc., Tokyo, Japan). The distinct wear track and many abrasion particles were generated on the surface of a silicon wafer substrate and in the surrounding areas, respectively. Although the three wear tracks have similar shapes, the degree of abrasion particle generation differs. The most abrasion particles were generated around the wear track produced by the custom-built micro-tribotester. The friction coefficient shows a different tendency; as wear occurs on the surface by contact sliding, a greater value of friction force is measured. However, the tendency of wear is enough to cause a slight difference even if the experiment is performed under the same conditions using the same device. The wear tracks generated by the three different tribotesters have similar wear mechanisms and the shape and degree of wear are very similar, thus verifying the reliability of the test results.

### 3.2. Tip Design Considering the Coupling Effect

In order to measure the friction coefficient using a strain gauge-based load sensor, the friction coefficient can be obtained by dividing each friction force by the normal force by measuring the normal and friction forces in real time simultaneously with two sensors. Alternatively, the friction coefficient can be obtained by applying the normal force by the dead weight method and measuring the friction force in real time through the load sensor, then dividing the friction force by the normal force of the same value. However, in the case of the strain gauge-based load sensor as described above, when the normal force is applied, the strain gauge of the load sensor to measure the friction force generated in the lateral direction is pressed in the vertical direction, so that the friction force might be measured through the strain gauge in the deformed state. In experiments with large friction force changes, it can be analyzed without major errors, but in the case of initial friction forces or small changes in the friction force, it is difficult to distinguish the error values that are incorrectly measured by the strain gauge. In this study, it is suggested to obtain the friction coefficient by measuring the change in the normal and friction forces using a fiberoptic sensor and cantilever tip. The cantilever tip of the tribotester based on a fiberoptic sensor has no strain gauge. Therefore, an erroneous signal caused by the deformation of strain gauge can be prevented. However, the deformation at the cantilever tip in the vertical direction occurs when the normal force is applied to the tip, which affects the measurement of the friction force. Accordingly, there is a relationship between the normal and friction forces depending on the degree of the deformation, stiffness, shape, and dimensions of the cantilever tip (Figure 7), as confirmed by simulation analysis. Tip 1 is the same tip used in the experiment comparing the three different tribotesters mentioned above. Tip 2 is smaller than Tip 1 and the shape of the cantilever was changed. Tips 2 and 3 have the same width as Tip 1 (25 mm), but their length is different, that is, 26 mm, and 36 mm, respectively. The thickness of Tip 2 was decreased to 0.3 mm; the thickness of Tip 3 is 0.65 mm, that is, the same as that of Tip 1. The three different tips are all made of stainless steel. The structures of Tips 2 and 3 with the similar shape are more deformable under small forces compared with Tip 1. In other words, the width of both arms of the cantilever tip of Tip 1 that is directly connected to the main body is 0.35 mm, while the widths of Tips 2 and 3 were decreased to 0.3 mm and they are connected by four arms (with an arm on both sides). The amount of the displacement of the cantilever tips 2 and 3 is greater, even under a relatively small force. This structure leads to a small stiffness value based on which smaller displacements and force changes can be measured.

The three different cantilever tips were analyzed using FEA simulations. The displacement during which the end of the tip changes was measured while a constant load (normal force/friction force) was applied to the end of the tip in the vertical and lateral directions. The degree of deformation obtained through simulation was within the displacement range that can be measured using a fiberoptic sensor. The data reliability was proven using the calibration experiment regarding Tip 1. In this process, the influence of the shape and dimensions of various tips was not analyzed experimentally, but through FEA simulation. Normal/friction forces of 0.1, 0.5, 1.0, 2.0, and 3.0 mN were applied to the end of the tips and the displacement of the tip end in the vertical and lateral directions was measured (Appendix A). The displacement of Tip 3 (25 µm) is more than three times smaller than that of Tip 2 (85 µm) when a normal force of 3 mN is applied, while the displacement of Tip 3 (13 µm) is approximately two times larger than that of Tip 2 (6.8) when a frictional force of 3 mN is applied. The relationship between the force and displacement regarding the vertical and lateral directions can be adjusted by changing the shape and dimension of the tip. The stiffness values in the vertical and lateral directions of each tip were obtained from the simulations. The following order was obtained for the vertical stiffness value affected by the normal force: Tip 1 (710 N/m) > Tip 3 (120 N/m) > Tip 2 (35 N/m). The lateral stiffness value in case of a friction force has the following order: Tip 1 (3950 N/m) > Tip 2 (440 N/m) > Tip 3 (235 N/m). It can be concluded that the cantilever tip suitable for the friction coefficient measurement using the fiberoptic sensor–cantilever tip-based tribotester is Tip 3 because it has the minimum lateral stiffness, making measurements of minute changes in the friction force feasible. This tip also has a relatively larger vertical stiffness and thus the coupling effect between the normal and friction forces can be reduced. In addition, the pressure generated at the cantilever tip during the application of normal and friction forces was analyzed (Appendix A). The maximum pressure was observed at the edge parts of the arms of the cantilever tip when the forces were applied. The maximum pressures generated in Tips 2 and 3 were similar when the friction force was applied in the lateral direction; however, the maximum pressure at Tip 3 was three times smaller than that of Tip 2 when the normal force was applied in the vertical direction. According to the shape and dimensions of the cantilever tip, when the displacement occurs in the vertical direction, it is confirmed that the magnitude of the generated pressure varies at the edge parts of the arms the deformed tip, that is, at the point where the maximum pressure occurs. A smaller maximum pressure, which is applied to the edge parts of both arms, was obtained for Tip 3, indicating that Tip 3 is the appropriated design, facilitating a reduction of the coupling effect when the normal force is applied.

The coupling effect based on the normal and friction forces was evaluated regarding Tip 3, which was selected based on the FEA simulation calibration regarding the force-displacement relationship. In other words, the coupling effect of the normal and friction forces was evaluated by comparing the tip deformation when only one normal or friction force was generated with that when the normal/friction forces were simultaneously applied in the vertical and lateral directions. The coupling effect represents the displacement change of the cantilever tip that occurs when normal and friction forces were simultaneously applied in the vertical and lateral directions, respectively. The lateral displacements of Tip 3 were measured when friction forces of 0.1, 0.5, 1.0, 2.0, and 3.0 mN and normal forces of 0.1, 0.5, 1.0, 2.0, and 3.0 mN were simultaneously applied (Appendix A). The results show that the magnitude of the maximum pressure at the edge parts of the arms of Tip 3 is significantly affected by the normal force compared with the friction force (Appendix A). The maximum pressure at the edge parts of the arms of the cantilever tip was greater when the normal and friction forces were simultaneously applied. However, the difference in the lateral displacement was insignificant when a normal force was applied to each friction force and when only the friction force was applied. The friction force-displacement relationship in the situation where the force is not applied in the vertical direction but only in the lateral direction is used as a comparison group. The stiffness value of the cantilever tip depending on the friction force regarding each normal force was calculated to be 234.8–235.2 N/m. All stiffness values are similar. Based on these results, it can be concluded that the coupling effect on Tip 3 is insignificant. The coupling effect of both forces must be regarded when the shape or size of a cantilever tip are changed depending on the range of the load conditions and predicted experimental friction forces. This can be verified by relatively simple FEA simulations, that is, without actually making various types of cantilever tips and/or conducting a lot of experiments.

## 4. Conclusions

A tribotester equipped with a fiberoptic sensor and cantilever tip was designed and produced and its performance as an experimental device that can be used to evaluate friction and wear properties was verified. Based on experiments and FEA simulation analyses, the degree of the deformation of the cantilever tip in the direction of normal/friction forces was measured with fiberoptic sensors. First, the calibration on the displacement was performed. Subsequently, the force-displacement calibration was conducted by measuring the stiffness of the cantilever tip and matching the forces to each displacement. A friction experiment was conducted in relative sliding reciprocating motion using this tribotester based on the fiberoptic sensor with a cantilever tip. The reliability of the device was proven by comparing the results obtained for the friction coefficient and wear track with those obtained by other tribotesters equipped with strain gauge-based load sensors. Cantilever tips with various shapes and sizes were designed, a normal/friction forces–tip displacement calibration was conducted, and the coupling effect based on normal/friction forces was verified to be negligible. Since the problems related to strain gauge-based load sensors could be solved by using the tribotester equipped with the fiberoptic sensor and cantilever tip, which measures the normal and friction forces without a strain gauge, as suggested in this research, the utilization of this design for the evaluation of tribological properties in various fields is expected.

## Figures and Tables

**Figure 1 micromachines-10-00748-f001:**
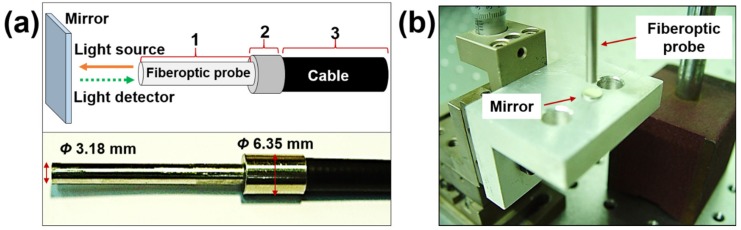
Principle of the distance measurement of the fiberoptic sensor: (**a**) schematic design and photo; (**b**) setup used for the calibration of the sensing distance measurement.

**Figure 2 micromachines-10-00748-f002:**
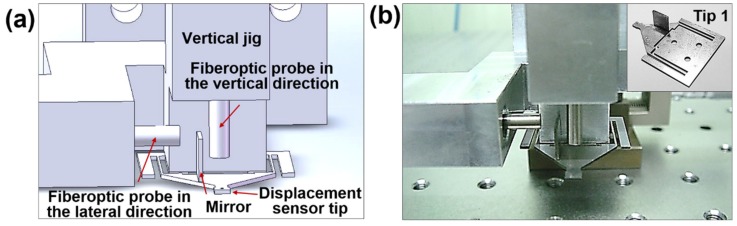
Experimental setup: (**a**) schematic design and (**b**) photo of the fiberoptic sensor–jig set and cantilever tip. Inset image (**b**): cantilever tip.

**Figure 3 micromachines-10-00748-f003:**
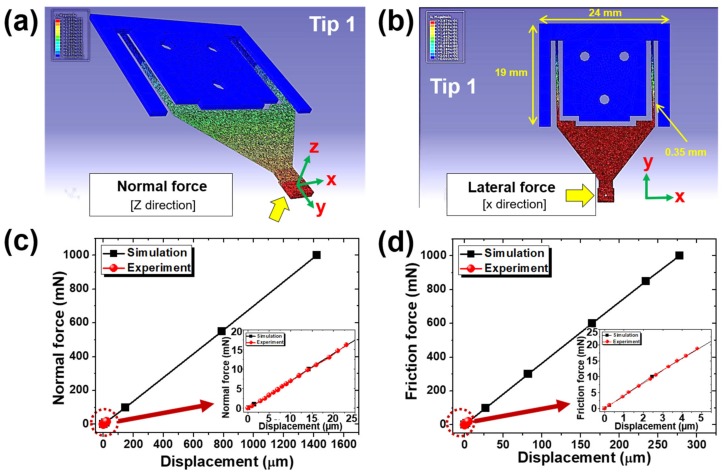
Calibration of the cantilever tip using finite element analysis (FEA) simulations: (**a**,**b**) results of FEA simulation showing the displacement of a cantilever tip in (**a**) vertical and (**b**) lateral directions by a force of 1 N; (**c**,**d**) comparison of relationship between (**c**) normal/(**d**) friction forces and displacements in vertical/lateral directions by experiment and FEA simulation.

**Figure 4 micromachines-10-00748-f004:**
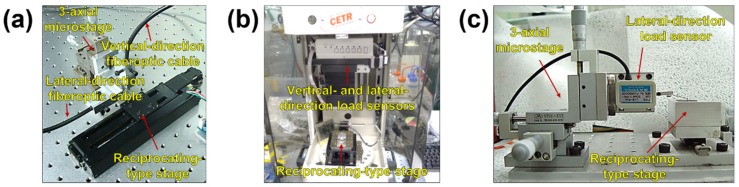
Images of (**a**) custom-built tribotester with two fiberoptic sensors and a cantilever tip; (**b**) commercial tribotester (CETR) consisting of two load sensors; (**c**) Custom-built micro-tribotester comprising a load sensor.

**Figure 5 micromachines-10-00748-f005:**
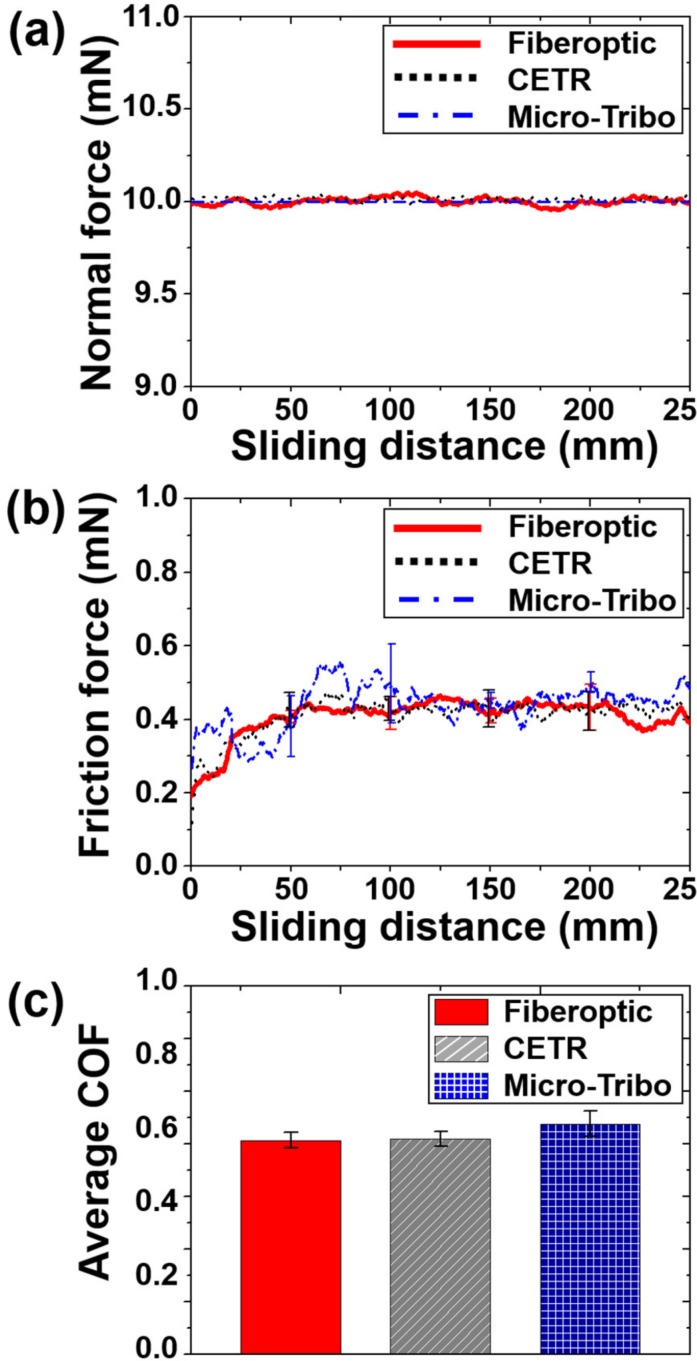
Variation of the (**a**) normal force and (**b**) friction coefficient with respect to the sliding distance; (**c**) average friction coefficient (COF) measured using the cantilever tip of the fiberoptic sensor-cantilever tip-based tribotester, commercial (CETR) tribotester, and custom-built micro-tribotester.

**Figure 6 micromachines-10-00748-f006:**
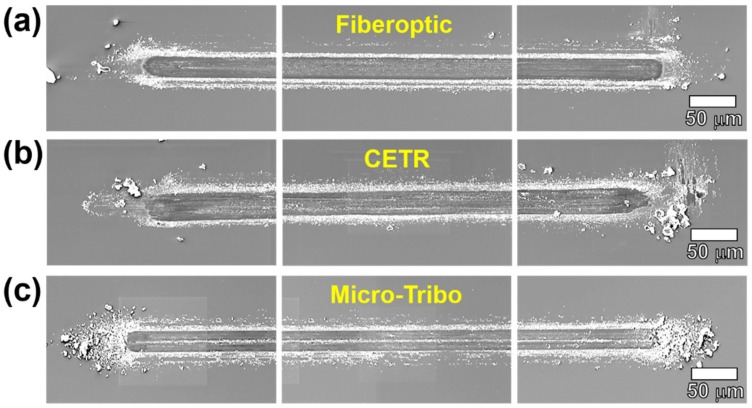
SEM images of the wear tracks on the Si wafer substrate generated by zirconia balls using the (**a**) fiberoptic sensor–cantilever tip-based tribotester, (**b**) commercial (CETR) tribotester, and (**c**) custom-built micro-tribotester.

**Figure 7 micromachines-10-00748-f007:**
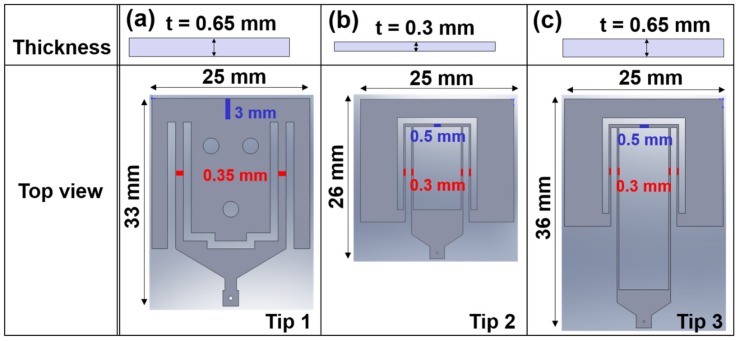
Schematic designs of the cantilever tips with different shapes and sizes. (**a**) Tip 1; (**b**) Tip 2; (**c**) Tip 3. Top: Thickness, Bottom: Top view.

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
