# Peer review of "Design of a Tribotester Based on Non-Contact Displacement Measurements"

_micromachines, 2019, doi:10.3390/mi10110748_

Round 1

Reviewer 1 Report

The paper presents a design of tribotester based on optical sensors. Overall, the design and validation with commercial product are interesting, however it appears that it is out of the scope of the journal: a) Fundamentals and Physics, b) Micromachines in Biology, c) Micromachines in Chemistry, d) Materials and Processing.

Besides, it is very difficult to read - the written English should be improved.

Reviewer 2 Report

I think the paper is well written and the materials covered are interesting for the tribology community. 

I have been trying hard to make some constructive suggestions and comments about this work but I have failed simply because the thoughtfulness of the author.

Therefore, in my opinion, the paper deserves to be published without any significant change.

Reviewer 3 Report

In general the paper is well written.

The figures in the paper could be improved. They are really difficult to read. In general they should be magnified.

Detailed comments to figures.

Figure 1: Picture c and d do not really tell anything except that there is a linear relationship between physical displacement and measured displacement. You can leave them out.

Figure 2: You can take away picture a and magnify the two other pictures b and c.

Figure 3: Things look very linear, so there is no reason to show this figure.

Figure 4: Picture a does not show a normal force acting on the tip. Make a projection picture similar to picture b and show the force acting there.

Figure 5: All pictures should be magnified significantly.

Figure 6: The figure is relevant, but should be magnified. Otherwise it is impossible to read anything from the figure.

Figure 9: There is no relevant information in this figure.

Figure 10: I don't see the point with figure 10.

Table 1: I don't see the point with table 1.
